# Distributed electrified heating for efficient hydrogen production

Hanmin Yang[1], Ilman Nuran Zaini[1], Ruming Pan[2], Yanghao Jin [1], Yazhe Wang [1], Lengwan Li[3], José Juan Bolívar Caballero[1], Ziyi Shi[1], Yaprak Subasi[4], Anissa Nurdiawati[5], Shule Wang [6,7], Yazhou Shen[8], Tianxiang Wang[9], Yue Wang[9], Linda Sandström [10], Pär G. Jönsson[1], Weihong Yang[1] & Tong Han [1] ✉

This study introduces a distributed electrified heating approach that is able to innovate chemical engineering involving endothermic reactions. It enables rapid and uniform heating of gaseous reactants, facilitating efficient conversion and high product selectivity at specific equilibrium. Demonstrated in catalyst-free $CH_4$ pyrolysis, this approach achieves stable production of $H_2$ (530 g h$^{-1}$ L$_{reactor}$$^{-1}$) and carbon nanotube/fibers through 100% conversion of high-throughput $CH_4$ at 1150 °C, surpassing the results obtained from many complex metal catalysts and high-temperature technologies. Additionally, in catalytic $CH_4$ dry reforming, the distributed electrified heating using metallic monolith with unmodified Ni/MgO catalyst washcoat showcased excellent $CH_4$ and $CO_2$ conversion rates, and syngas production capacity. This innovative heating approach eliminates the need for elongated reactor tubes and external furnaces, promising an energy-concentrated and ultra-compact reactor design significantly smaller than traditional industrial systems, marking a significant advance towards more sustainable and efficient chemical engineering society.

Currently, the chemical industry continues to depend on conventional chemical engineering techniques involving strong endothermic chemical reactions[1,2]. Reaction rate and chemical equilibrium exponentially depend on temperature for these reactions[3]. To attain chemical equilibrium swiftly, all reactants must reach a specific threshold temperature quickly and uniformly and receive a prompt energy supply for reactions to occur. However, in modern reactors, the energy supplied for heating reactants and reactions relies on the heat transfer from an external heated surface to the interior of reactants that flow[4]. Although

using a catalyst can lower the requirements for reaction temperature and the corresponding reaction energy[5], the low thermal conductivity of reactants (especially gaseous reactants) and the endothermic reactions lead to non-uniform temperature distribution, exhibiting significant temperature gradients across the reactor (Gas temperature profile in Fig. 1A)[6,7]. To achieve a higher conversion rate at the outlet of industrial reactors with high-throughput gas feeding, many elongated reactor tubes with a high length-to-diameter ratio are necessary[8,9]. However, equipping many elongated reactor tubes with external

[1]Department of Materials Science and Engineering, KTH Royal Institute of Technology, SE-10044 Stockholm, Sweden. [2]School of Energy Science and Engineering, Harbin Institute of Technology, Harbin 150001, China. [3]Department of Fiber and Polymer Technology, Wallenberg Wood Science Center, KTH Royal Institute of Technology, Stockholm SE-100 44, Sweden. [4]Department of Chemistry - Ångström Laboratory, Structural Chemistry, Uppsala University, Lägerhyddsvägen 1, 751 21 Uppsala, Sweden. [5]Department of Industrial Economics and Management, KTH Royal Institute of Technology, 10044 Stockholm, Sweden. [6]International Innovation Center for Forest Chemicals and Materials, College of Chemical Engineering, Nanjing Forestry University, Longpan Road 159, Nanjing 210037, China. [7]Jiangsu Province Key Laboratory of Biomass Energy and Materials, Institute of Chemical Industry of Forest Products, Chinese Academy of Forestry (CAF), No. 16, Suojin Five Village, Nanjing 210042, China. [8]Department of Mechanical Engineering, Imperial College London, London SW7 2AZ, UK. [9]Department of Civil and Architectural Engineering, KTH Royal Institute of Technology, SE-100 44, Sweden. [10]Department of Biorefinery and Energy, RISE Research Institutes of Sweden AB, Box 726, SE-941 28 Piteå, Sweden. ✉e-mail: tongh@kth.se

combustion furnaces results in bulky reaction systems (Fig. 1A). Taking commercialized catalytic steam methane reforming (SMR) as an example, the large industrial reformer system (>1000 m³) consists of an array of more than 100 tubular reactors 10 cm to 17 cm in diameter and 10 m to 12 m in length (length to diameter ratio >50)[10]. In addition, dynamic changes in reactants' temperature along the reactor lead to shifts in chemical equilibrium[3,11]. This, in turn, leads to decreased selectivity of desired products, which is expensive to separate to obtain high-purity products[3,12].

Numerous efforts have been made towards the electrification of chemical engineering processes, which provides a viable solution for energy storage and the integration of renewable energy sources[13–15]. Additionally, this approach holds the promise of eliminating the need for external combustion furnaces, which in turn results in a more energy-efficient and low-carbon method[6,16–18]. However, in most electrified research, the energy needed for heating the reactants and reactions is still transferred from an externally heated surface to the interior of flowing reactants, resulting in elongated reactor tubes[6,17]. Combining the use of a structured catalyst with electrification is considered a further advancement, as the structured catalyst can enhance the heat and mass transport phenomena, ensuring a flat temperature profile along the entire catalytic bed[16,19,20].

This study presents a distributed electrified heating method to tackle the existing challenges. By distributing gaseous reactants through numerous electrically heated regular micron-scale channels, the method allows for rapid and uniform heating of high-throughput gaseous reactants to a precise reaction temperature with minimal temperature gradients and immediate energy supply to prompt the strong endothermic reactions (gas temperature profile in Fig. 1B). This results in highly efficient and selective production of desired products by maintaining constant chemical equilibrium of all reactants. Distributed electrified heating eliminates the need for a large number of elongated reactor tubes and external heating furnaces, leading to an energy-concentrated and ultra-compact reactor system (Fig. 1B). We use distributed electrified heating to conduct two endothermic reactions, i.e., CH₄ pyrolysis (Eq. (1)) and CH₄ dry reforming (Eq. (2)), to showcase the disruptive advantages of using this method on green H₂ and syngas production.

$$CH_4 \leftrightarrow C_S + 2H_2, \Delta H^0_{25°C} = 74.8\ kJ/mol \qquad (1)$$

$$CH_4 + CO_2 \leftrightarrow 2CO + 2H_2, \Delta H^0_{25°C} = 247\ kJ/mol \qquad (2)$$

## Results

### Efficient high-throughput CH₄ pyrolysis through a metal-free wood carbon monolith

As illustrated in Fig. 2A, by simply using a metal-free 3D wood carbon monolith, our study achieves stable pyrolysis of CH₄ and produces high-purity H₂ and carbon nanofibers/tubes. The carbon monolith used in this study has dimensions of approximately 2 cm * 2 cm * 2 cm and is made from natural spruce wood through pyrolysis and subsequent carbonization (Figs. S1, S2). It features open, elongated, and tortuous channels with diameters of 10–60 μm in the axial direction (Fig. S3) and secondary mesopores (2–25 nm) and micropores (0.5–2 nm) on the channel walls (Fig. S4, Supplementary Discussion 1). Induction heating of the carbon monolith allows for electrical heating of all walls of the micron-scale channels (Figs. 2A, S5), enabling immediate energy supply to CH₄ in each channel, resulting in rapid and complete CH₄ decomposition at a constant temperature and corresponding chemical equilibrium (Fig. 2A).

As shown in Fig. S6, at a specific gas hourly space velocity (GHSV) of 750 h⁻¹, we observe CH₄ conversions of approximately 60%, 70%, 80%, and 100% at temperatures of 850 °C, 950 °C, 1050 °C, and 1150 °C, respectively (Table S2). Based on the tests conducted, it appears that the CH₄ conversions achieved through distributed electrified heating are almost equivalent to the CH₄ equilibrium conversions[21]. This indicates that the method enables all methane molecules to rapidly reach equilibrium without any heat transfer limitation. Moreover, as shown in Fig. S7, at 1150 °C, 100% CH₄ conversion is still observed at elevated GHSV of 1500 h⁻¹, 3000 h⁻¹, and 6000 h⁻¹. Despite a significant increase in gas flux, the maintenance of CH₄ conversion at equilibrium conversion (100%) is ensured, highlighting the distinct benefits of using distributed electrified heating for high-throughput gas feeding. Under GHSV conditions other than the GHSV of 6000 h⁻¹, H₂ is the only detected gas compound (H₂ concentration of 100%). Even if the GHSV is 6000 h⁻¹, the H₂ concentration is also higher than 94.5%. The key to successful technology implementation is maintaining stable testing performance over an extended period. A 1200-min test of CH₄ pyrolysis (first round for 720 min and second round for 480 min) was

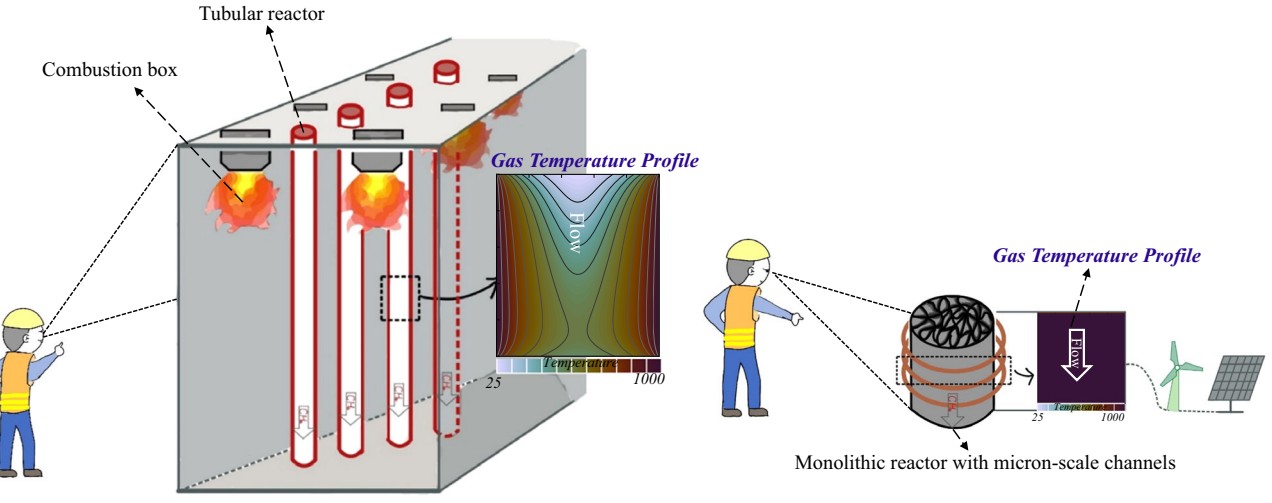

**A. Conventional Heating**   **B. Distributed Electrified Heating**

**Fig. 1 | Conventional heating reactor system VS Distributed electrified heating reactor system. A** Conventional heating reactor system; (**B**) Distributed electrified heating reactor system in this study. Source data are provided as a Source Data file.

conudcted at 1150 °C and GHSV of 3000 h⁻¹ (Fig. 2B). CH₄ conversion and H₂ concentration close to 100% were observed in the long-term test, with only minor fluctuations due to varying testing conditions. The corresponding H₂ production capacity is approximately 530 g h⁻¹ L$_{reactor}$⁻¹. At the end of the test, there is no noticeable decrease in CH₄ conversion and H₂ concentration. This study demonstrates stable pure H₂ (without CO₂ or other byproducts) production through ~100% CH₄ conversion, at a temperature of 1150 °C and high-throughput GHSV of 3000 h⁻¹ without the assistance of metal catalysts[22,23]. Moreover, all tests in this work use pure CH₄ without carrier gas as the reactor feed, enabling on-site pure H₂ production without separation and making it ideal for applications that require high H₂ purity, such as fuel cells[24]. A comparative study shows that distributed electrified heating outperforms most ongoing developed technologies using complex metal catalysts or fultra-high temperature in terms of process simplicity, stability, CH₄ conversion, H₂ concentration, gas flux (GHSV), and reaction temperature (Table S3).

In terms of CH₄ pyrolysis reaction, its potential of solid carbons production is just as significant as H₂ production, making it a valuable market application. The capacity for solid carbon production is approximately 1590 g h⁻¹ L$_{reactor}$⁻¹, which realize almost complete conversion of carbon elements in CH₄ into the solid carbon. Distributed electrified heating produces solid carbons in the form of curled fibrous carbons (Fig. 2C), as opposed to the typical spherical carbon blacks produced through direct CH₄ pyrolysis in the absence of a metal catalyst[25]. These fibrous carbons have diameters that fall within two size ranges: 10–100 nm and 300–1000 nm (Fig. 2C). Upon examining the transmission electron microscopy (TEM) images, it is apparent that fibrous carbons show a combination of constructions, some with hollows resembling bamboo-like nanotubes and some

without hollows resembling nanofibers (Fig. 2C). Carbon nanofibers and carbon nanotubes, which have greater commercial value than carbon blacks, are commonly formed with the aid of a metal catalyst[26]. The extensive production of carbon nanofibers and carbon nanotubes by using a metal-free wood carbon monolith and the high purity of the carbon products (no risk of metal contamination) indicate another disruptive advantages of the distributed electrified heating (Table S3). It can be inferred that the concentrated localized heating effect of the induction heating facilitates the accumulation of carbon atoms into fibers (Figs. S10–S13, Table S4, and Supplementary Discussion 2). In addition, it is confirmed that solid carbons can be quickly carried out from the carbon channels during testing, ensuring a continuous process (Figs. S14 and S15, Table S1, and Supplementary Discussion 3).

Even though the wood carbon monolithic reactor can function reliably for a certain period, it will eventually reach the end of its life cycle. As part of our research, an electrochemical study is conducted to assess the effectiveness of utilizing certain spent wood carbon monoliths (specifically, those that had undergone a 1200-min stability test) as anodes for sodium-ion batteries (SIBs). The prepared SIB half-cell demonstrated an initial coulombic efficiency (ICE) value of 94.7% and maintained a decent revisable capacity of 218.9 mAh/g (Fig. 2D). These values are much higher than that demonstrated by the fresh carbon monolith (Fig. 2D, Fig. S16). The stability of the electrode material was evident from the 20-cycle charge-discharge curve (Fig. 2E, Fig. S17). These findings suggested that the spent wood carbon monolith could be a viable option to be used as SIB anode materials. The deposition of extra carbon nanotubes/fibers onto the porous surface of wood carbon effectively closes the open pores, resulting in a significant increase in the ICE for SIBs (Fig. S18, Supplementary Discussion 4).

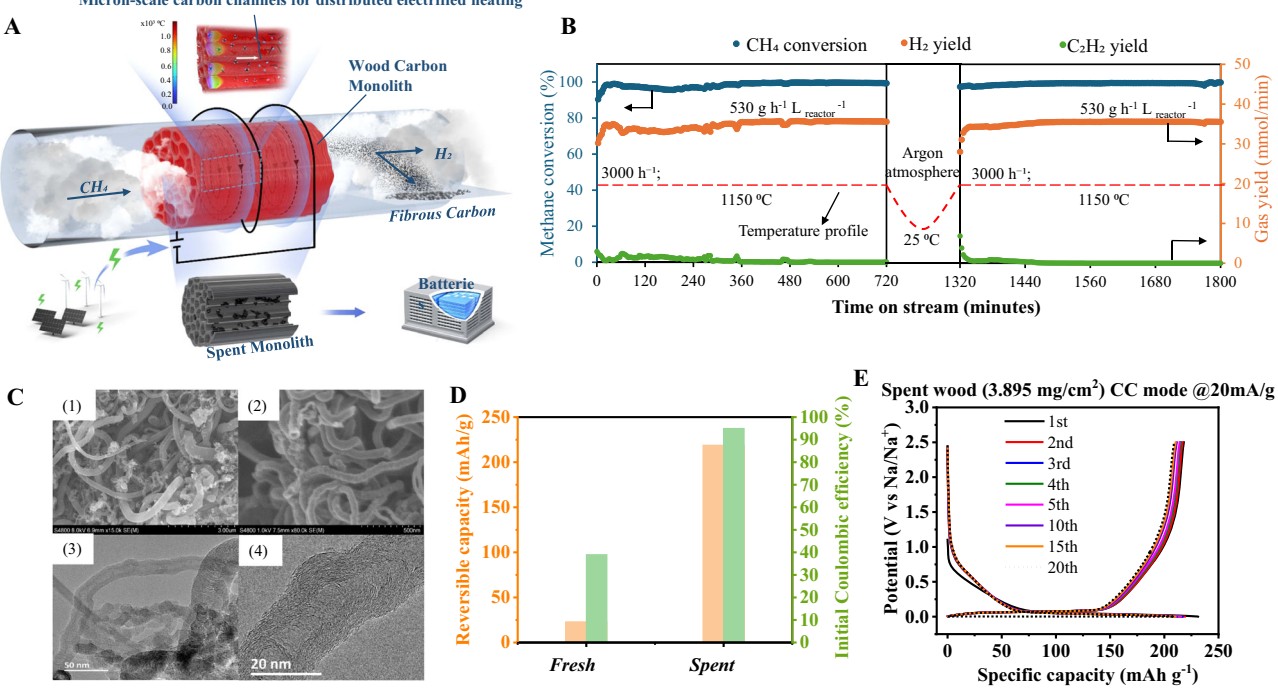

**Fig. 2 | CH₄ pyrolysis using the distributed electrified heating method.**
**A** Schematic diagram of the CH₄ pyrolysis process (Using a wood carbon monolith with micron-scale channels able to be induction heated, CH₄ pyrolysis produces pure H₂ and carbon nanotubes/fibers. The spent monolith can also be used as a battery anode material.); (**B**) Long-term stability test of CH₄ pyrolysis at 1150 °C and GHSV of 3000 h⁻¹; (**C**) SEM and TEM images of the carbon nanotubes/fibers: C-1 and C-2 SEM images of carbon nanotubes/fibers under different resolutions, C-3 and C-4 TEM image of carbon nanotubes/fibers with combined constructions; (**D**) Electrochemical performance comparison of fresh and spent wood carbon monolith samples; (**E**) Cycling performance of the spent wood carbon monolith sample. CC mode: Constant Current charing mode. Source data are provided as a Source Data file.

## Efficient catalytic $CH_4$ dry reforming with high syngas production capacity

Distributed electrified heating is also applicable for endothermic reactions involving a catalyst. As depicted in Fig. 3A, catalytic dry reforming of $CH_4$ over a common Ni/MgO catalyst is carried out in a metallic monolith reactor with equilateral triangle channels of 300 μm side length. Distributed electrified heating is accomplished through induction heating of the metal monolith, where all channel walls are electrically heated up to ensure immediate energy supply to reactants (Figs. S19 and S20).

In Fig. 3B, the performance of catalytic dry reforming of $CH_4$ is displayed for different temperatures and weight hour space velocity (WHSV). When the WHSV is 75.8 L $g_{cat}^{-1}$ $h^{-1}$, a conversion rate of almost 100% is observed for both $CH_4$ and $CO_2$ at temperatures of 750 °C, 800 °C, and 850 °C. The syngas product has a fixed $H_2$ to CO ratio of approximately 1, indicating a high desired product selectivity. Figure S21 shows the equilibrium performance of the $CH_4$ dry reforming reaction (without catalyst). Our study shows higher $CH_4$ and $CO_2$ conversion at lower temperatures mainly attributed to the implementation of the Ni/MgO catalyst[27]. For the 30.8 mL reactor (with a diameter of 3.5 cm and a length of 3.2 cm) used in this study, the total syngas production capacity can reach a value of 47.4 L $h^{-1}$ under these conditions (Fig. 3C). In terms of Ni quantity, the capacity for producing $H_2$ and CO is approximately 7.3 mol per gram of Ni per hour (Fig. 3C). Even when the WHSV is at 113.7 L $g_{cat}^{-1}$ $h^{-1}$ and 151.6 L $g_{cat}^{-1}$ $h^{-1}$, a conversion rate of 100% for $CO_2$ is still maintained, while a conversion rate of approximately 90% is observed for $CH_4$. Correspondingly, the

syngas product has a decreased $H_2$ to CO ratio of approximately 0.9. The syngas production capacity of the reactor increases to 63.2 L $h^{-1}$ (WHSV of 113.7 L $g_{cat}^{-1}$ $h^{-1}$) and 82.5 L $h^{-1}$ (WHSV of 151.6 L $g_{cat}^{-1}$ $h^{-1}$). The capacity for $H_2$ production is increased to 9.5 mol $g_{Ni}^{-1}$ $L^{-1}$ (WHSV of 113.7 L $g_{cat}^{-1}$ $h^{-1}$) and 12 mol $g_{Ni}^{-1}$ $L^{-1}$ (WHSV of 151.6 L $g_{cat}^{-1}$ $h^{-1}$). The capacity for CO production is increased to 10.4 mol $g_{Ni}^{-1}$ $L^{-1}$ (WHSV of 113.7 L $g_{cat}^{-1}$ $h^{-1}$) and 13.7 mol $g_{Ni}^{-1}$ $L^{-1}$ (WHSV of 151.6 L $g_{cat}^{-1}$ $h^{-1}$).

We compare our testing performance with other literature reports that involve modified Ni/MgO catalysts showing optimized performance or the implementation of the same induction heating method[28–32]. As shown in Fig. 3D, the common Ni/MgO catalyst used in this study outperforms most optimized catalysts in terms of $CH_4$ conversion (Fig. 3D (1)), $CO_2$ conversion (Fig. 3D (2)), syngas production capacity (Fig. 3D (3)), and $H_2$ to CO ratio (Fig. 3D (4)) with the assistance of distributed electrified heating, even without any modification. To achieve close to 100% conversion of $CH_4$ and $CO_2$, our distributed electrified heating-powered Ni/MgO catalyst requires the lowest temperature of 750 °C. In addition, the $H_2$ to CO ratio is maintained at 1 (Fig. 3D (4)), indicating that side reactions are minimized and the reaction equilibrium remains consistent. More importantly, the reactor demonstrates this exceptional activity performance, with a syngas production capacity tens of times higher than other studies (Fig. 3D (3)). When the conversion rate and product selectivity of a reaction are comparable, a higher syngas production capacity of a reactor means a higher reactant throughput and a correspondingly higher energy supply requirement. All these emphasize the advantages of our distributed

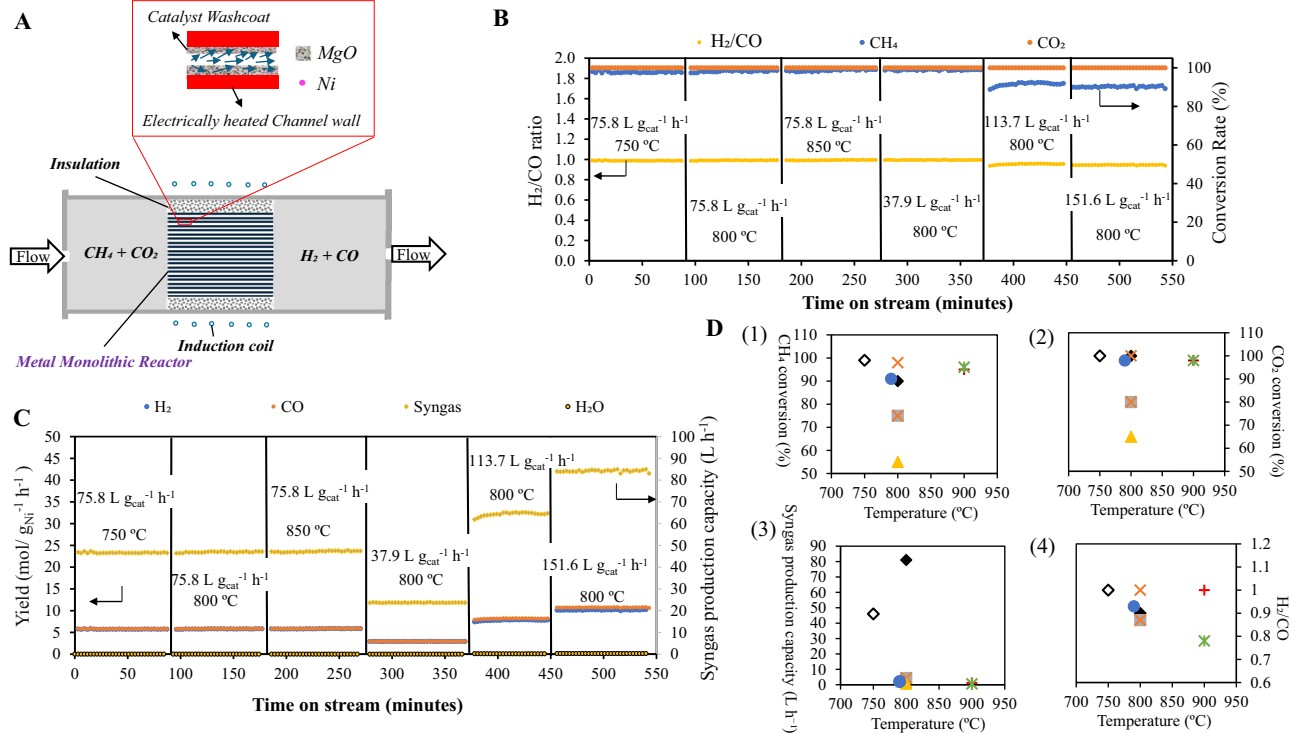

**Fig. 3 | Catalytic $CH_4$ dry reforming using the distributed electrified heating method. A** Schematic diagram of the catalytic $CH_4$ dry reforming process (Using a metallic monolith with a common Ni/MgO washcoat, distributed-electrified-heat $CH_4$ dry reforming produces high purity $H_2$ and CO syngas with a $H_2$/CO ratio close to1); (**B**) Performance of the catalytic $CH_4$ dry reforming at various conditions; (**C**) Yield of $H_2$ and CO, and syngas production capacity of the reactor at various conditions. The production capacity of syngas is determined by summarizing the volumes of $H_2$ and CO produced; (**D**) Distributed-electrified-heat $CH_4$ dry reforming over common Ni/MgO catalyst show disruptive advantages to the literature reports using optimized Ni/MgO in terms of $CH_4$ conversion (D-1), $CO_2$ conversion

(D-2), syngas production capacity (D-3) and $H_2$/CO ratio (D-4): ◇ this study, 750 °C, 7% Ni/MgO, WHSV of 75.6 L g $cat^{-1}$ $h^{-1}$; ◆ this study, 800 °C, 7% Ni/MgO, WHSV of 151.6 L g $cat^{-1}$ $h^{-1}$; ✕ ref. 28, 800 °C, 3.76% Ni- 1.76% Mo/MgO (single crystal), WHSV of 60 L g $cat^{-1}$ $h^{-1}$; ▩ ref. 28, 800 °C, 3.76% Ni- 1.76% Mo/MgO (single crystal), WHSV of 360 L g $cat^{-1}$ $h^{-1}$; ▲ ref. 29, 800 °C, NiCo alloy pellets powered by induction heating; ● ref. 30, 790 °C, 20% NiO/Nano MgO; ✕ ref. 31, 900 °C, 1%Ni/ Mg0.85Zr0.15 O; ✚ ref 32, 900 °C, 10% Ni/CeO2-MgO. Source data are provided as a Source Data file.

electrified heating method. The literature's induction heating method failed to perform well possibly due to the lack of regular micron-scale channels and thereby a uniform and rapid heating of reactants and immediate energy supply to reactions cannot be achieved[29]. After testing for a total of 540 min, 461.01 L of syngas are produced, and there was no decrease in catalyst activity associated with coke formation throughout the test. This suggest that the advanced heating method could also aid in maintaining catalyst stability within the investigated reaction time. Moving forward, the combination of distributed electrified heating and an optimal catalyst composition (showing optimized performance with long stability) is the way forward for the large-scale application of the catalytic CH$_4$ dry reforming reaction.

### Ultra-fast and uniform heating of reactants with minimal temperature gradients

To better understand the experimental results, we used computational fluid dynamics simulations to illustrate the advantages of the distributed electrified heating method for heating gaseous reactants. In this study, the simulation uses CH$_4$ as the reactant gas. We calculate the temperature distribution of gases within the reactor for two scenarios using different heating methods (Table S6): external heating and distributed electrified heating.

As illustrated in Fig. 4A, the temperature contours of the external heating scenario conform to the heat transfer characteristics from the tube wall to the interior of the flowing gas. Substantial temperature difference along the radial position of the reactor is observed throughout the reactor. Temperatures of the gases adjacent to the reactor wall (radial positions of 0 and 2 cm) are close to the preset temperature i.e., 1150 °C. Nevertheless, temperatures of the gases in the center (radial position of 1 cm) are significantly low (approximately 50 °C at the gas inlet and approximately 600 °C at the gas outlet). On the contrary, a uniform temperature distribution is

found for the distributed electrified heating scenario (Fig. 4B). As the gas flows along the axial direction, there is a rapid increase in temperature, reaching a stable point of 1150 °C. The gases coming in are distributed into numerous micron-scale channels, each with walls that are electrically heated to a fixed temperature of 1150 °C. The gas flow temperature increases rapidly to 1150 °C in the axial direction within a single channel (from 0 to 0.05 cm, as illustrated in Fig. 4B for a single channel), resulting in rapid and uniform heating of the entire gas flow. Figure 4C compares the temperature gradient in a radial direction across the reactor under two different heating methods. Correspondingly, the traditional external heating scenario shows a relatively high-temperature gradient in the radial direction along the whole reactor. Furthermore, it can be observed that with an increase in radial position, there is a significant decrease in temperature gradient (Fig. 4C). This trend is also evident in the calculated temperature gradients in the axial direction (Fig. S23). These findings indicated that the conventional external heating method has major limitations when it comes to heat transfer among gases, resulting in non-uniform and slow heating. In the distributed electrified heating scenario, the temperature gradient in the radial direction throughout the reactor is consistently close to 0, which aligns with its flat temperature contours. There is an intense temperature gradient in the axial direction near the gas inlet that rapidly decreases to a value of approximately zero (Fig. S23), demonstrating the advantage of utilizing the induction heating method for rapid and uniform heating. It is worth noting that the gas velocity in the distributed electrified heating scenario (up to 0.7 m/s, Fig. S24) is significantly higher than in the external heating scenario (up to 0.16 m/s, Fig. S25) due to the rapid heating and uniform temperature distribution. When gas flows are faster, it helps to remove carbon more quickly, thereby significantly reducing the risk of carbon blockages in the channels. This explained the steady progress of CH$_4$ pyrolysis to some extent.

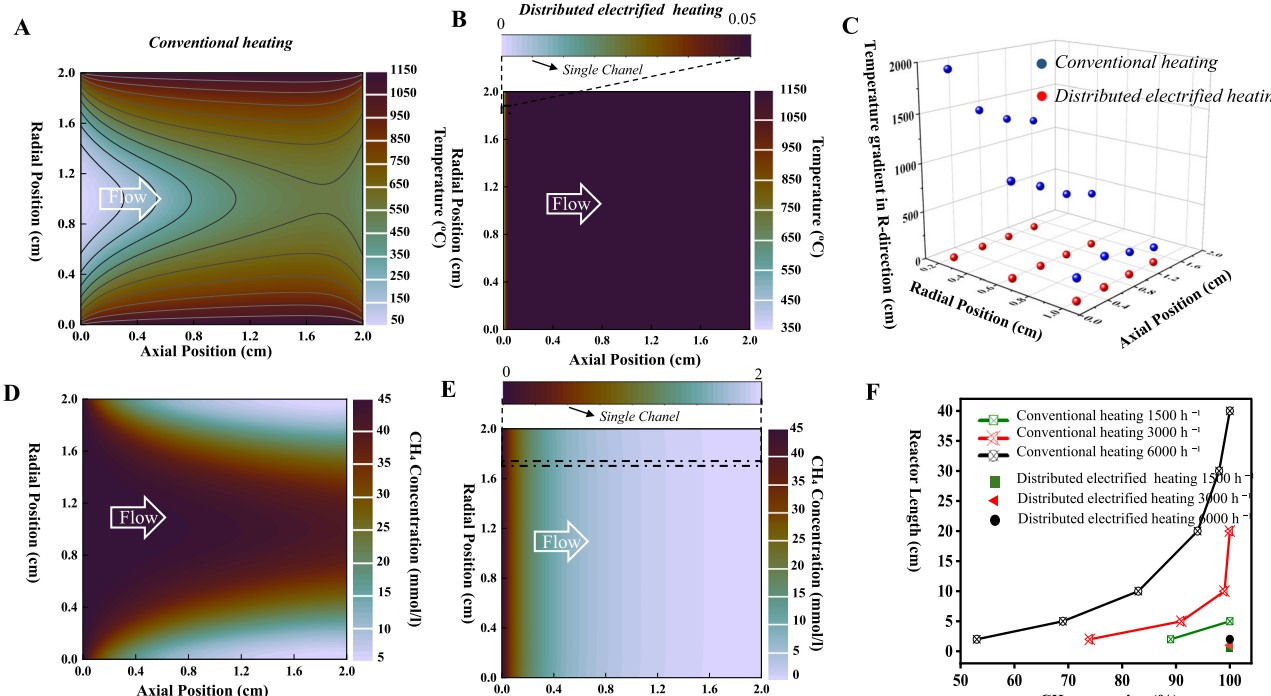

**Fig. 4 | Computational fluid dynamics (CFD) calculations. A** Gas temperature distribution under conventional heating; (**B**) Gas temperature distribution under distributed electrified heating; (**C**) Calculated temperature gradients in radical direction of two different heating methods; (**D**) CH$_4$ concentration distribution under conventional heating; (**E**) CH$_4$ concentration distribution under distributed electrified heating; (**F**) Required reactor length to reach certain conversion of two heating method with a same reactor diameter of 2 cm. Source data are provided as a Source Data file.

## Efficient reaction in an energy-concentrated and ultra-compact reactor system

The computational model also integrates reaction kinetics to determine $CH_4$ pyrolysis conversion rate in real-time. A steep difference in $CH_4$ conversion is observed across the reactor for external heating (Fig. 4D), while fast $CH_4$ conversion is observed along the axial direction for distributed electrified heating (Fig. 4E). The accuracy of the kinetic data simulation is verified by comparing it to measured data at the reactor outlet. The temperature plays a crucial role in determining the equilibrium conversion of $CH_4$. In contrast to traditional external heating, where heat transfer severely restricts the reaction. Distributed electrified heating remains the reaction at a steady state due to the highly consistent temperature, promoting rapid $CH_4$ pyrolysis and preventing the occurrence of side reactions[3].

Fig. 4F compares the reactor length required for certain $CH_4$ conversions based on the simulation at elevated GHSV for two different heating scenarios. For a fixed reactor diameter of 2 cm, a reactor length of 2 cm is enough to achieve 100% $CH_4$ conversion at all GHSVs for the distributed electrified heating case, verified by the experimental result. In the case of external heating, the reactor length needs to be extended to 40 cm, which is 20 times longer, to achieve a 100% conversion of $CH_4$ at a GHSV of 6000 $h^{-1}$. In addition, if the diameter of the reactor is increased, it will require a substantial increase in its length as the limitation of heat transfer will significantly increase[4,7]. However, this concern can be avoided through the use of distributed electrified heating. Literature reports have shown that eliminating the requirement for an externally fired furnace is expected to reduce the reactor's volume by approximately 100 times[6]. With our distributed electrified heating, external combustion furnaces are eliminated as well. On this basis, the length of the reaction tube can be shortened by over 20 times. Therefore, using distributed electrified heating is possible to achieve a volume reduction of over 2000 times when compared with an industrial tubular reactor with a side-fired furnace (Fig. 5). Based on the images of the reactor from different angles during the test (Fig. S27), it is evident that our distributed electrified heating is highly effective in concentrating heat in the reaction areas. In contrast to conventional external heating, the temperature inside the reactor is significantly higher than that of the insulation and tube walls (Fig. 5). This enables the energy to be used for gas heating and endothermic reactions to a greater extent, conforming to a highly energy-concentrated and ultra-compact system. This technology innovates conventional chemical processes, towards compact reactor design and energy saving. Especially for $CH_4$ pyrolysis, the ultra-compact reactor and its ability to produce high-purity $H_2$ without requiring separation makes it ideal for supplying on-site $H_2$ for fuel cells in electric vehicles. It presents a potential solution to the predicament of $H_2$ storage in the transportation industry[33,34].

## Techno-economic analysis

Based on the experimental data, we have designed a process model to conduct a techno-economic analysis (TEA) of the $CH_4$ pyrolysis process using ASPEN PLUS V12.1 (Table S7). The proposed system, illustrated in Fig. 6A, comprises a primary induction-heat $CH_4$ pyrolysis reactor with regular micron-scale channels, a cyclone for separating the carbon product, a heat recovery system, a multi-stage compressor, and a pressure swing adsorption (PSA) unit. The primary feedstock used is natural gas. The gas composition is based on the typical composition of natural gas in the European market[35]. Overall, the process model design converts 100 kg of natural gas into 71.3 kg of carbon product (carbon nanofibers and carbon nanotubes), 22.0 kg of compressed $H_2$, and 6.7 kg of tail-gas (Fig. 6A). For the induction-heating reactor, an energy efficiency value of approximately 88.3% was used for the analysis[36]. The specific energy consumption for $H_2$ production is calculated to be approximately 1.28 kWh/Nm$^3$-$H_2$. 70% of the tail gas is recirculated to the $CH_4$ pyrolysis reactor to achieve the minimized

specific energy consumption (kWh/Nm$^3$-$H_2$) for the production of $H_2$ and carbon products (figure S28). To put this into perspective, we have compared it with the lowest specific energy consumption values for $H_2$ production in other research that uses electrified technologies. The comparison is shown in Fig. 6B. Our findings show that the specific energy consumption for $H_2$ production in this work is comparable with similar works, and in some cases even much lower than other electrified heating technologies such as joule-heated steam $CH_4$ reforming (SMR)[20], microwave-heated SMR[16], joule-heated dry $CH_4$ reforming (DMR)[37], microwave-heated DMR[38], and thermal plasma $CH_4$ pyrolysis[39].

The TEA of the process model is calculated for 100 kg/h of natural gas feeding. The cost of feedstock, specifically natural gas, is the largest contributor to the annual expenses of the process model (Tables S9 and 10). As shown in figure S29, natural gas expenses make up 50.7 % of the total expense, followed by capital expenditures (CAPEX, 25%), electricity (12.2%), and fixed operating expenditures (OPEX, 12.1%). In this study, the key economic metric employed is the levelized cost of hydrogen (LCOH), which is a measure of the average cost for producing hydrogen over the system's lifetime. It was calculated by determining the ratio of the total discounted costs for the life of the system to the total discounted hydrogen production, considering a lifetime of 25 years and a discount rate of 8%. The process of $CH_4$ pyrolysis produces two valuable products: $H_2$ and carbon nanofibers/tubes. Market analysis indicates that carbon nanofibers are priced between 20 EUR/kg to over 100 EUR/kg[40]. Therefore, in this study, the LCOH is calculated based on the different prices of carbon products. Figure 6C shows the calculated LCOH at different natural gas prices when the carbon product prices are 0.25 EUR/kg, 2 EUR/kg, 5 EUR/kg, 10 EUR/kg, and 20 EUR/kg, respectively. The electricity price is fixed as 80 EUR/MWh, representing the average electricity price of Europe in 2022[41]. When the carbon product is priced at 0.25 EUR/kg, the LCOHs calculated at a natural gas price of 80 EUR/MWh are 8.38 EUR/kg. The results show that producing only $H_2$ is not cost-effective for $CH_4$ pyrolysis processes in Europe, where electricity and natural gas prices are both high. Lowering the price of natural gas to 20 EUR/MWh results in a decrease of LCOH to 4.85 EUR/kg at a carbon product price of 0.25 EUR/kg. This indicates a significant impact caused by the natural gas price. At this low natural gas price, the LCOH can be reduced to €1.5/kg, the existing market price of hydrogen, by selling the carbon product at €1.28/kg. It is important to highlight that when the carbon product is priced at 5 EUR/kg or higher, the LCOH becomes negative, even under conditions where natural gas prices are significantly high at around 120 EUR/MWh (Fig. 6C). Considering the current market price of carbon nanofiber (between 20 €/kg to over 100 €/kg), the overall process is believed to be highly profitable. The $CH_4$ pyrolysis reactor using distributed electrified heating technology is the biggest source of price uncertainty in the model, as it is a new and commercially unavailable technology. A sensitivity analysis is conducted on the reactor CAPEX and the results are displayed in Fig. 6D. The natural gas and electricity prices have been fixed at 80 EUR/MWh. It is observed that decreasing or increasing the reactor CAPEX by 50% does not result in significant changes in the calculated LCOH values. Moreover, the LCOH values also become negative when the carbon product price reaches 5 EUR/kg. Table S11 presents a comparison of the LCOH calculated in this study with other $CH_4$ pyrolysis processes such as plasma and catalytic molten pyrolysis technologies. The calculated LCOH is comparable to other technologies at a carbon product price of 2 EUR/kg, which is a reference price for commercial carbon black. It's worth noting that the study uses the highest natural gas price of 80 EUR/kWh. Furthermore, our study is the only one that shows negative LCOH values because of the formation of carbon nanofibers/tubes that have a relatively high market price[40]. To sum up, TEA predicts a highly profitable $CH_4$ pyrolysis process that co-produces $H_2$ and carbon nanofibers/tubes by using distributed electrified heating technology.

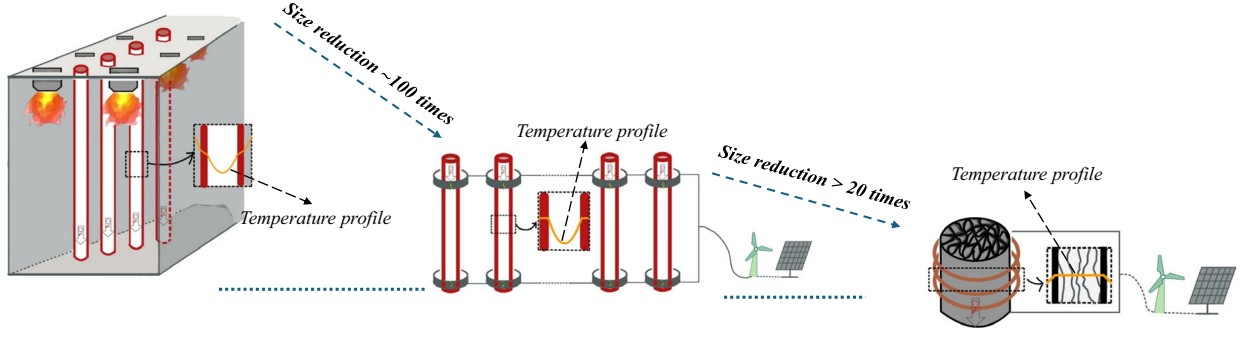

**Fig. 5 | An ultra-compact reactor system.** Volume reduction (~100 times) by an electrical tubular reactor compared to the conventional tubular reactor is mainly achieved by eliminating the external combustion furnace (ref. 6). The figure displays temperature profiles that depict the temperature contours of a specific horizontal location in reactors that employ various heating techniques.

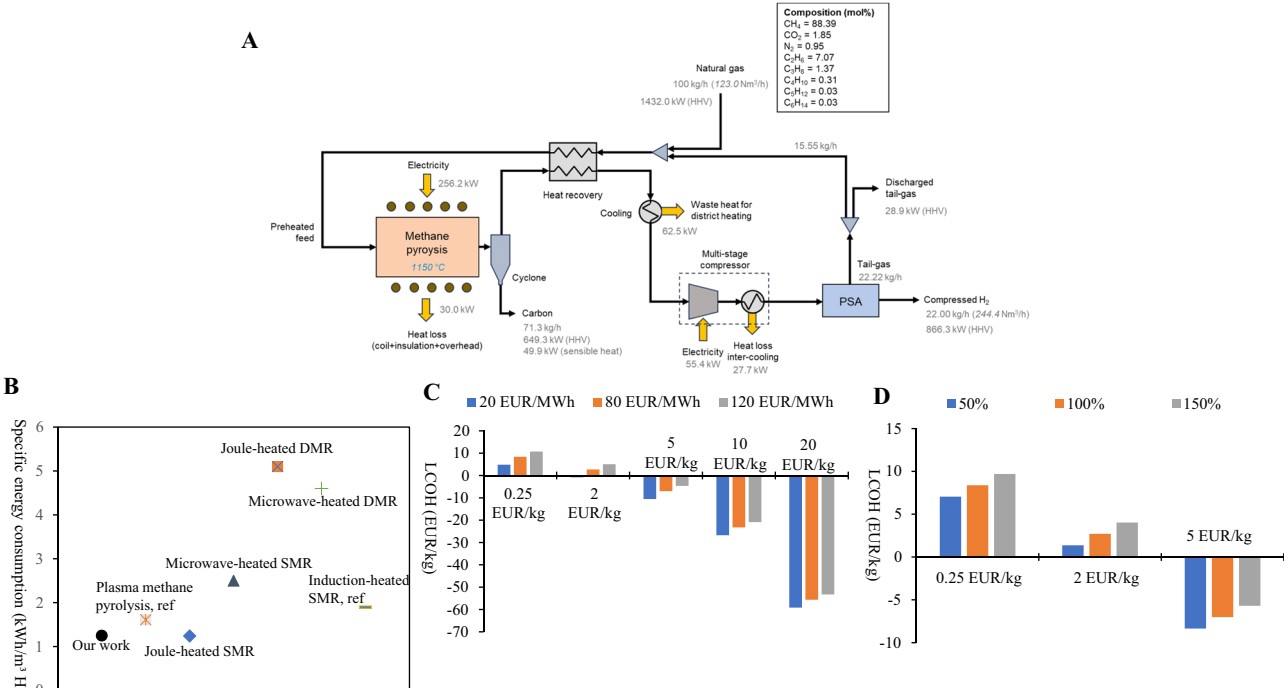

**Fig. 6 | Techno-economic analysis of methane pyrolysis in the distributed electrified heating process.** (**A**) The mass and energy flow of the distributed-electrified-heat $CH_4$ pyrolysis routine; (**B**) Comparison between specific energy consumption value for $H_2$ production in this work and the selected lowest specific energy consumption values for $H_2$ production in other research using electrified technologies; (DMR: dry methane reforming; SMR: steam methane reformring); (**C**) Calculated levelized costs of hydrogen (LCOH) values as a function of natural gas prices (20, 80, 120 EUR/MWh) at different carbon product selling price (0.25, 2, 5, 10, 20 EUR/kg); (**D**) Calculated LCOH values as a function of CAPEX of pyrolysis reactor (50%: 0.30 M EUR; 100%: 0.59 M EUR; 150%: 0.89 M EUR) at different carbon product selling price (0.25, 2, 5 EUR/kg). Source data are provided as a Source Data file.

## Discussion

This research highlights the benefits of utilizing distributed electrified heating in endothermic reactions. Model reactions for concept proof include catalyst-free $CH_4$ pyrolysis and catalytic $CH_4$ dry reforming. For $CH_4$ pyrolysis, stable production of pure $H_2$ and pure carbon nanotube/fibers through 100% conversion of $CH_4$ in the absence of metal catalysts is achieved at a low temperature of 1150 °C and high-throughput GHSV of 3000 h$^{-1}$. Using just a piece of carbon, distributed electrified heating achieves performance and stability beyond what many complex metal catalysts and ultra-high temperature heating technoogies can do. For catalytic $CH_4$ dry reforming, the reactor with a washcoat of common Ni/MgO catalyst without any modification achieves exceptional activity performance at a syngas production capacity tens of times higher than other studies using precisely modified Ni/MgO-based catalysts.

As opposed to conventional heating, distributed electrified heating enables rapid and uniform heating and reactions with minimal temperature gradients, maintaining at a consistent reaction chemical equilibrium. The conventional industrial tubular reactor is now history with the elimination of its external heating furnace and elongated reactor tube, resulting in a potential volume reduction of over 2000 times (Fig. 5). By incorporating renewable electricity, the technology

could play a vital role in expediting the journey of chemical industries toward carbon neutrality. Moreover, the techno-economic analysis predicts a highly profitable $CH_4$ pyrolysis process that co-produces both $H_2$ and carbon nanofibers/tubes by using distributed electrified heating technology.

## Methods

### $CH_4$ pyrolysis using wood carbon monolith reactor
Carbon monoliths fabricated by pyrolysis and subsequent carbonization of Norwegian spruce wooden cubes were used in this study for the $CH_4$ pyrolysis reaction. The wooden cubes were loaded into a horizontal furnace and heated at a rate of 1 °C/min until reaching 800 °C, after which they were held at that temperature for 3 h in an $N_2$ atmosphere.

During the $CH_4$ pyrolysis test, a PHILIPS induction heating furnace-based system was used (figure S5). The test involved inserting the insulation material and carbon monolith into a vertical quartz tube measuring 50 mm in diameter and 500 mm in length. The quartz tube was then connected to the gas supply and product gas analysis instrument through flanges at each end. Prior to the test, the incoming $CH_4$ or Ar gas was regulated by mass flow controllers calibrated through an electrical soap film flowmeter. Gas analysis was performed using a gas chromatograph (µGC, 490 Micro-GC System QUAD, Agilent) with four columns - Molsieve 5 Å, PoraPLOT U, $Al_2O_3$/KCl, and CP-Sil 5CB. The testing phase began with elevated temperatures of 850 °C, 950 °C, 1050 °C, and 1150 °C, at a fixed $CH_4$ flow rate of 100 mL/min (corresponding to a GHSV of 750 $h^{-1}$). The GHSV is computed by dividing the $CH_4$ flow rate by the volume of the carbon monolith. After it, the test was changed to the elevated flow rates of 100 ml/min (750 $h^{-1}$), 200 ml/min (1500 $h^{-1}$), 400 ml/min (3000 $h^{-1}$), and 800 ml/min (6000 $h^{-1}$), at a fixed temperature of 1150 °C by using a new wood carbon monolith. For each test condition, the reaction persists for approximately 30 min. The stability of the test was also conducted at 1150 °C and 400 ml/min (3000 $h^{-1}$) by using a new wood carbon monolith. After the test, solid carbons were collected from the wood carbon monolith's top surface and the connecting flanges' top cooling area. For a more detailed understanding of the matter, kindly refer to section 1.1 in the Supplementary Information.

### Catalytic $CH_4$ dry reforming using a metal monolith reactor
A 32 mm long FeCrAl alloy-based monolith with a diameter of 35 mm and a cell density of 200 cpsi is used for catalytic $CH_4$ dry reforming test. Before the test, the monolith is coated with Ni/MgO catalyst by Hulteberg Chemistry & Engineering AB (https://www.hulteberg.com/). The resulting coating is approximately 0.18 g NiO/1.89 g MgO on the metallic monolith, equaling to a Ni loading of approximately 7%.

The reactor system consists of a horizontal tube (made from quartz tube) with the coated metallic monolith is placed in the middle of the tube (figure S19). An induction heater is used to heat the metallic monolith. Prior to the test, the metallic monolith reactor was heated to a temperature of 500 °C and a flow of 750 mL/min 4%-$H_2$/$N_2$ for 8 h to undergo reduction. The reactor is then heated to the specific testing temperature according to the test plan. After the temperature reaches the point, 8%-$CH_4$/$N_2$ mixture and $CO_2$ gases are injected into the metallic monolith reactor after passing through a gas mixture. During the test, the ratio of $CH_4$ to $CO_2$ is fixed at a value of 1. The gas flow rate is set according to WHSV values used during the test. Specifically, for a WHSV of 75.8 L $g_{cat}^{-1}$ $h^{-1}$, the flow rate of $CH_4$/$N_2$ mixture and $CO_2$ is set as around 2500 and 200 ml $min^{-1}$. Syngas producing from the metallic monolith reactor is then cooled and sent to the micro-GC for an online monitoring followed by a gas meter for the volume determination.

### Sample characterization
Texture properties of the fresh and spent wood carbon monolith are determined by obtaining $N_2$ and $CO_2$ adsorption–desorption isotherms obtained at 77 K and 298 K by using a Micromeritics model ASAP 2020 instrument. SEM observation is carried out by using a JEOL JSM-7800F instrument (20 kV and 10 mm working distance) instrument equipped with a Bruker AXS XFlash Detector 4010 (MA, USA). EDS detector from Oxford Instrument is further equipped on the SEM instrument to perform the samples' elemental composition and surface mapping analysis. TEM was performed at room temperature on a JEOL JEM-2100 microscope equipped with a LaB6 gun operated at 200 kV. Eurofins Biofuel & Energy Testing Sweden AB (https://www. eurofins.com/) conducted ultimate elemental analysis and the corresponding ash content and composition analysis of the wood carbon monolith sample. Raman spectra were obtained by using a Tyrode I Raman microscope equipped with a 532-nm wavelength diode laser. For a more detailed understanding of the matter, kindly refer to section 2 in the Supplementary Information.

### Electrochemical performance test
In order to evaluate the effectiveness of spent wood carbon monolith as anodes for SIBs, pouch-type half-cell batteries are assembled and tested. To evaluate the electrochemical performance, the cell is subjected to galvanostatic cycling using a LAND potentiostat instrument. This is carried out in CC mode, at a current of 20 mA/g, within the voltage range of 0.001 to 2.5 V, and at a temperature of 25 °C. For a more detailed understanding, please refer to section 3 in the supplementary information.

### Computational fluid dynamics simulations
The model was implemented in COMSOL Multiphysics 6.0 in a 2D-axisymmetric geometry with fully coupled equations for fluid motion, energy transport, and mass transport. The simulation uses $CH_4$ as the reactant gas for pyrolysis reactions. The reactor model is simplified by treating it as a porous medium with pore size and porosity similar to the wood carbon monolith. External and induction heating are simulated as boundary and domain heating of the porous media, respectively. The simulation of electromagnet induction heating is simplified by setting the solid temperature at a constant value that is achieved by the experiments. Heat is derived from the boundary with a fixed temperature for boundary heating, and heat is derived from the solid porous medium (skeleton) with a fixed temperature for domain heating. The primary focus of this research is to examine the variations in gas reactant heating caused by two heating methods. Kindly refer to section 4 in the Supplementary Information for more details.

### Process simulation and techno-economic analysis
A system for co-producing hydrogen and carbon is simulated based on a distributed electrified methane pyrolysis reactor that has been investigated. The proposed system, illustrated in Fig. 6A, comprises a primary pyrolysis reactor heated through induction, a cyclone for separating the carbon product, a heat recovery system, a multi-stage compressor, and a PSA unit. The primary feedstock used was natural gas with a typical composition from the European market[35]. The process model was created using ASPEN PLUS V12.1 (Aspen Technology, Inc) to conduct heat and mass balance calculations, taking into account thermodynamic equilibrium for the methane pyrolysis process. Main process parameters and assumptions can be found in Supplementary Information (Table S7). To optimize $H_2$ and carbon production at a minimum specific electricity consumption (kWh/$Nm^3$-$H_2$), a portion of the tail gas discharged from the PSA is recirculated to the pyrolysis reactor. Through sensitivity analysis, it was determined that the minimum specific electricity consumption is attained with a 70% tail-gas recycle (Fig. S28).

Techno-economic assessment (TEA) has been conducted in this study, and the key economic indicator employed is the Levelized Cost of Hydrogen (LCOH), which is a measure of the average cost to produce a unit of hydrogen over the lifetime of a project or facility. The

calculation of the LCOH, capital expenditure (CAPEX), and operating expenditure (OPEX) are illustrated in the Supplementary Information.

## Reporting summary

Further information on research design is available in the Nature Portfolio Reporting Summary linked to this article.

## Data availability

The experiment and simulation data generated in this study are provided in the Source Data file. Source data are provided with this paper.

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

## Acknowledgements

This study was financial supported by VINNOVA- the Swedish innovation fund Agency (2021-03735, W.Y.), and Energimyndigheten- the Swedish Energy Agency (51418-1, W.Y.). One of the authors, Hanmin Yang would also like to acknowledge the financial support from the Chinese Scholarship Council (CSC) and Stiftelsen Energitekniskt Centrum i Piteå, Sweden. Open access funding provided by KTH Royal Institute of Technology.

## Author contributions

H.Y., T.H., I.N.Z. and W.Y. conceived the idea and designed the research. H.Y., I.N.Z., Y.J., Ya.W., J.J.B.C. and T.H. fabricated the reactor and conducted the tests. R.P. and Y.Sh. performed the computational fluid dynamics simulation. Y.J., L.L., Z.S. and Y.Su. carried out the characterization of materials and electrochemical measurements. I.N.Z., A.N. and S.W. simulated the process modeling and performed the techno-economic assessment. T.W. and Yu.W. provided the wood materials and conducted the mechanical measurement. H.Y., I.N.Z. and T.H. analyzed the data, designed the figures, and wrote the first draft of the manuscript. L.S., P.G.J. and W.Y. supervised the project. All authors have revised and approved the manuscript.

## Funding

## Competing interests

The authors declare no competing interests.
