## [Peer Review File · Nature Communications]

REVIEWER COMMENTS

Reviewer #1 (Remarks to the Author):

In this paper, the authors proposed the use of distributed electrified heating method for performing different chemical processes.

The topic is very interesting and actual, but the paper needs to be carefully revised and improved before being considered for publication in the journal.

- The structure of the paper needs to be revised: the reviewer sees only the section abstract. Where are, for example, the sections "introduction" and "conclusions"? Moreover, in the abstract the references to the literature must not be present;
- The authors should increase the literature survey by adding more recent papers. Some examples of papers to add are: 10.1016/j.renene.2023.04.082, 10.1016/j.renene.2022.07.157, 10.3390/en15103588;
- Equations 1 and 2 are very difficult to read;
- Perhaps some material and methods detail present in the supplementary information could be moved to the main text;
- The authors should better specify in which way they calculated the GHSV;
- The authors should add the thermodynamic equilibrium data in figure 3;
- The authors should add more details in the description of the preparation of the monolithic catalyst: for example which is the calcination temperature?
- Which is the role of steam (figure S18)?
- Which is the energy consumption of the proposed induction heating technique? Can the authors compare this value with the one of some other electrified H₂ production processes in literature?
- Did the authors perform any mesh optimization in the CFD simulations?
- The authors wrongly referred to figure S17 at line 263: did they mean figure S25?
- The authors should better comment figure 3C: how is the syngas curve (the yellow one, I suppose) obtained? Is it the sum of produced H₂ and CO?

Reviewer #2 (Remarks to the Author):

Reviewer's comments on manuscript: "Distributed electrified heating: a potential way to revolutionize modern chemical engineering"

Manuscript ID: NCOMMS-23-41087-T

General comment: This paper reports on an electricity-based and inductive heating method that attains rapid and uniform heating of gaseous reactants at reaction temperature suitable for highly efficient reactants conversion and high desired product selectivity at constant chemical equilibrium. The method is tested for highly endothermic reactions i.e., CH₄ pyrolysis and dry reforming of CH₄.

The topic of CH₄ valorisation powered by electricity is always of high interest. The work has been meticulously performed and analysed. The reaction experiments are convincingly supported by the respective material characterization and CFD simulations. The supplementary information adequately supports the research findings and discussion. Nevertheless, the general concept is not new, to the best of my knowledge. Other prominent works have already been published on inductive-based heating for CH₄ valorisation i.e., doi.org/10.1016/j.cattod.2019.05.005; doi.org/10.1016/j.ijhydene.2020.09.262; doi.org/10.1016/j.ijhydene.2019.02.055 and doi.org/10.1016/j.cej.2021.132934. I really like the idea of the piece of wood carbon use to promote CH₄ pyrolysis and the use of the spent monolith as a battery anode material but those elements are not sufficient to justify a publication in the Nature communication Journal. I would encourage the authors to clearly state in the abstract i) the novelty of the present work; ii) how this work differentiates from the previously published works and iii) how the findings go beyond the state-of-the-art. In my opinion journal such as international journal of hydrogen energy, chemical engineering and processing: process intensification and chemical engineering science would be more suitable for this work.

Some specific comments which the authors may take into account to improve the quality of the present manuscript are listed below:

Line 32: You shall replace "conversation" to "conversion".

Line 57-58: "Though using a catalyst can lower this requirement..." ❓ Which requirement are you referring to?

Line 95-96: "...complete CH₄ decomposition at a constant temperature and corresponding chemical equilibrium (Fig 2A)." and Line 101: "...electrified heating are almost equivalent to the CH₄ equilibrium conversions" ❓ Could you please add the respective equilibrium figure for CH₄ pyrolysis for the temperature range you achieve in this work?

Line 101-102: "This indicates that the method enables all reactants to rapidly reach equilibrium..." ❓ Why do you use plural (reactants)? There is only CH₄ fed into the reactor.

Line 104: You must refer to Figure S7. Please replace Figure S5 to Figure S7.

Line 104-106: “Despite a significant increase in gas flux, the maintenance of CH₄ conversion at equilibrium conversion (100%) is ensured, highlighting the distinct benefits of using distributed electrified heating for high-throughput gas feeding.” ☒ This is true only for the highest temperature, i.e., 1150°C. Why is this not pronounced at lower temperatures (Figure S6)? Please present the equilibrium conversion for this temperature range.

Line 107-108: “H₂ is the only detected gas compound (H₂ concentration of 100%).” ☒ At temperatures lower than 1150°C, other species (mainly C₂) are expected to be formed based on equilibrium composition presented in doi.org/10.1016/0378-3820(94)00109-7. Why do you detect only H₂ and carbon?

Line 127-128: “... which is 3 times higher than the capacity for H₂ production.” ☒ This is redundant; C=12 g/mol and H=1 g/mol ☒ based on the mass balance, the mass production of C will always be 4 times the production of H (12 vs 4*1 in CH₄ molecule).

Line 144-145: “Even though the wood carbon monolithic reactor can function reliably for a certain period, they will eventually reach the end of their life cycle.” ☒ What is happening to its composition and structure? So, a continuous process (according to the string definition) is not feasible! A cyclic operating mode is rather feasible. Please be more precise.

Line 148: SIBs ☒ Please explain the acronym.

Line 165-167: “WHSVs). When the WHSV is 75.8 L gcat⁻¹ h⁻¹, a conversion rate of almost 100% is observed for both CH₄ and CO₂ at temperatures of 750 °C, 800 °C, and 850 °C.” ☒ As previously asked, please present the equilibrium performance of dry methane reforming at this temperature range.

Line 172-173: “...a conversion rate of 100% for CO₂ is still maintained, while a conversion rate of approximately 90% is observed for CH₄.” ☒ 100% seems quite a lot. Please present the carbon lack in the conversion/temperature plots.

Line 182-185: “As shown in Figure 3D, the common Ni/MgO catalyst used in this study outperforms most optimized catalysts in terms of CH₄ conversion (Fig 3D-1), CO₂ conversion (Fig 3D-2), syngas production capacity (Fig 3D-3), and H₂ to CO ratio (Fig 3D-4) with the assistance of distributed electrified heating, even without any modification.” ☒ Why? What is done differently than other works? It is not clear to me what the difference is; is the design of the reactor, the type of the catalytic material, the structure of the catalytic material or something else?

Line 188: "...indicating that side reactions are..." ❑ Why are those side reactions? Could you also comment on the reaction mechanism?

Line 191-192: "A higher syngas production capacity of a reactor means a higher reactant throughput..." ❑ Not necessarily; it can also attributed to higher single-pass conversion. Please rephrase accordingly.

Line 193-195: "The literature's induction heating method failed to perform well possibly due to the lack of regular micron scale channels for uniform and rapid heating and immediate energy supply." ❑ Please add the respective reference(s).

Line 200-201: "... is the way forward for large-scale application of the catalytic CH₄ dry reforming reaction." ❑ This is generality. The authors do not explain how their reactor can be scaled. Please elaborate further.

Line 228-230: "In the distributed electrified heating scenario, the temperature gradient in the radial direction throughout the reactor is consistently close to 0, which aligns with its flat temperature contours." ❑ In my opinion, this is promoted by the small and confine geometry. What if the channels were wider, which might be the case for industrial scale reactors.

Line 237-238: "Perhaps the steady progress of CH₄ pyrolysis can be explained by this." ❑ What would be the reactor performance if conductive (Joule) heating would be applied? I believe the result would be the same. If so, does inductive heating bring an added value as compared to other electrified heating methods such as conductive heating? Could you please comment on this matter?

Line 267-268: "This enables the energy to be used for gas heating and endothermic reactions to a greater extent, conforming to a highly energy- concentrated and ultra-compact system." ❑ What is the global energy efficiency (electricity-to-chemical energy)?

Supplementary information material

Figure S6 ❑ Why does CH₄ conversion decrease over the time at 850, 950 and 1050 oC while at 1150 oC remains stable?

Table S3 I strongly recommend you to include an additional metric, that is the energy cost: actual energy provided vs theoretical energy needed to attain the same result for each case. This will allow fair comparison considering all important aspects.

Reviewer #3 (Remarks to the Author):

The work is interesting and useful and has the potential to be accepted, but some points must be clarified or fixed before I can proceed and positive action can be taken.

1- The abstract is long and should be more concise.

2- Why does the paper lack an Introduction section and a review of previous relevant studies? Also, there is no conclusion section.

3- In line 278, it is mentioned that 3000 h⁻¹ represents an industrial GHSV. The reference of this issue must be addressed.

4- According to the paper, mini reformers such as compact monolithic modules are good candidates to apply the approach of distributed electrical heating. This is while, the use of small-scale reactors that include micro-channels, on an industrial scale is associated with limitations. This problem is aggravated in the case of reactions that cause path closure by producing coke (such as methane pyrolysis); because decoking of micro-channels is a tedious attempt. This reduces the industrial attractiveness of these reactors and casts doubt on the authors' claim about the potential of creating an industrial revolution. If the authors believe that this method is only applicable to micro-scale reactors, they should clearly explain the limitations of using this method in industrial scales.

Reviewer #1 (Remarks to the Author):

In this paper, the authors proposed the use of a distributed electrified heating method for performing different chemical processes.

The topic is very interesting and actual, but the paper needs to be carefully revised and improved before being considered for publication in the journal.

- The structure of the paper needs to be revised: the reviewer sees only the section abstract. Where are, for example, the sections “introduction” and “conclusions”? Moreover, in the abstract the references to the literature must not be present;

Response:

Thanks for your comment. The structure of the paper has been revised accordingly. Sections titles including “introduction”, “conclusions”, and “methods” have been added. Furthermore, we have revised the abstract section by removing the background information and reference literature. Please see page 2 for the updated **Abstract** section.

- The authors should increase the literature survey by adding more recent papers. Some examples of papers to add are: [10.1016/j.renene.2023.04.082](https://doi.org/10.1016/j.renene.2023.04.082), [10.1016/j.renene.2022.07.157](https://doi.org/10.1016/j.renene.2022.07.157), [10.3390/en15103588](https://doi.org/10.3390/en15103588);

Response:

Thanks for your valuable comment. We have conducted a more detailed literature review and have incorporated the relevant information into the Introduction section of the manuscript. To support our description, we have also included recent research papers listed by the reviewer. Please refer to page 3-4, paragraph 2 to review the updated content in the Introduction section.

- Equations 1 and 2 are very difficult to read;

Response:

Thanks for your comment. Equations 1 and 2 have been revised for clarity. Please review the updated equations on page 4.

- Perhaps some material and methods detail present in the supplementary information could be moved to the main text;

Response:

Thanks for your comment. We have added the Methods section in the appropriate place of the manuscript. Please refer to the highlighted contents on pages 16-19.

- The authors should better specify in which way they calculated the GHSV;

Response:

Thanks for your comment. We have updated the “Methods” section of the main content (page 16, paragraph 2) and section “**1.1.2 CH₄ pyrolysis test**” in Supplementary Information (Page 7, red-marked sentence) to include the calculation of GHSV : “*The GHSV is determined by dividing the CH₄ flow rate by the volume of the carbon monolith.*” We appreciate your input and hope that this clarification is helpful.

- The authors should add the thermodynamic equilibrium data in figure 3;

Response:

Thanks for your comment. The thermodynamic equilibrium of the CH₄ dry reforming reaction has been added in the supplementary file. Please refer to Supplementary Information, page 44, figure S21.

- The authors should add more details in the description of the preparation of the monolithic catalyst: for example which is the calcination temperature?

Response:

Thanks for your comment. We have included additional information in the description of the preparation process for the coating of the metal monolith. This includes the calcination temperature, duration of heating, and the gas atmosphere used. These details have been added to provide better understanding about the preparation process. Please refer to section **1.2.1 Preparation of the metal monolith reactor with Ni/MgO catalyst washcoat** in Supplementary Information, page 10-11.

- Which is the role of steam (figure S18)?

Response:

Thank you for your comment. As stated in the manuscript, we did not use steam in our study actually. The steam stream depicted in the original figure S18 was solely included to demonstrate that the reactor system is capable of being used for steam-involved reactions, such as methane steam reforming, which is also a research interest for our group. To avoid confusion, figure S18 has been revised accordingly. In the revised file, it is figure S19. Kindly refer to the updated figure S19 in Supplementary Information, page 42.

- Which is the energy consumption of the proposed induction heating technique? Can the authors compare this value with the one of some other electrified H₂ production processes in literature?

Response:

Thanks for your comment. Based on our calculation in the TEA section, the specific energy consumption for H₂ production of the CH₄ pyrolysis process using the proposed distributed electrified heating technique is estimated to be around 1.28 kWh/m³ -H₂. To put this into perspective, we have compared it with the lowest specific energy consumption values for H₂ production in other research that uses electrified technologies. The comparison is shown in Fig 6B. Our findings show that the specific energy consumption for H₂ production in this work is

comparable with other similar works, and in some cases even much lower than other electrified heating technologies such as joule-heated steam CH₄ reforming (SMR), microwave-heated SMR, joule-heated dry CH₄ reforming (DMR), microwave-heated DMR, and thermal plasma CH₄ pyrolysis. Kindly refer to the section **Techno-economic analysis** in the manuscript, page 12-13.

- Did the authors perform any mesh optimization in the CFD simulations?

Thanks for your comment. The main objective of the current study is to propose the distributed electrified heating method and to demonstrate the disruptive advantages of using this method on green H₂ and syngas production. Two endothermic reactions, i.e., non-catalytic CH₄ pyrolysis and catalytic CH₄ dry reforming have been conducted as examples. The CFD simulations are mainly used to verify the advantage of the distributed electrified heating method compared to the conventional heating method. In the CFD simulations, we used a fixed geometry with a specific mesh without mesh optimization. We believe this is enough for the concept proof.

Mesh optimization is always essential for the implementation of the distributed electrified heating method. Actually, we are currently working on a project that combines 3D printing and electrified heating technology. The project involves optimizing the mesh using CFD simulations and constructing a monolith using 3D printing. The 3D-printed monolith is made of FeCrAl alloy, which can be heated using induction heating and direct electrical heating. A manuscript is under preparation, but unfortunately, we cannot share more information until it is published.

- The authors wrongly referred to figure S17 at line 263: did they mean figure S25?

Response:

Thanks for your comment. We apologize for incorrectly referencing figure S17 in line 263 of the manuscript. We have revised the content of manuscript, and after adjusting the figures sequence, the figure S25 has been renamed as figure S27. Please refer to the manuscript, page 12:

“Based on the images of the reactor from different angles during the test (figure S27), ...”

- The authors should better comment figure 3C: how is the syngas curve (the yellow one, I suppose) obtained? Is it the sum of produced H₂ and CO?

Response:

Thank you for your comment. To determine the production capacity of syngas, we calculated the combined volumes of H₂ and CO produced. We have revised the caption of Figure 3 to include this clarification. Please check the marked content in manuscript on page 26.

Reviewer #2 (Remarks to the Author):

Reviewer’s comments on manuscript: “Distributed electrified heating: a potential way to revolutionize modern chemical engineering” Manuscript ID: NCOMMS-23-41087-T

General comment: This paper reports on an electricity-based and inductive heating method that attains rapid and uniform heating of gaseous reactants at reaction temperature suitable for highly efficient reactants conversion and high desired product selectivity at constant chemical equilibrium. The method is tested for highly endothermic reactions i.e., CH₄ pyrolysis and dry reforming of CH₄. The topic of CH₄ valorisation powered by electricity is always of high interest. The work has been meticulously performed and analysed. The reaction experiments are convincingly supported by the respective material characterization and CFD simulations. The supplementary information adequately supports the research findings and discussion.

Nevertheless, the general concept is not new, to the best of my knowledge. Other prominent works have already been published on inductive-based heating for CH₄ valorisation i.e., doi.org/10.1016/j.cattod.2019.05.005; doi.org/10.1016/j.ijhydene.2020.09.262; doi.org/10.1016/j.ijhydene.2019.02.055 and doi.org/10.1016/j.cej.2021.132934. I really like the idea of the piece of wood carbon use to promote CH₄ pyrolysis and the use of the spent monolith as a battery anode material but those elements are not sufficient to justify a publication in the Nature communication Journal. I would encourage the authors to clearly state in the abstract i) the novelty of the present work; ii) how this work differentiates from the previously published works and iii) how the findings go beyond the state-of-the-art. In my opinion journal such as international journal of hydrogen energy, chemical engineering and processing: process intensification and chemical engineering science would be more suitable for this work. Some specific comments which the authors may take into account to improve the quality of the present manuscript are listed below:

Response:

Thanks for your comment. We are grateful for your positive feedback on the presentation of our article and the experiments, characterizations, and simulations involved. The major objective of our work is to propose a method of distributed electrified heating and demonstrate the disruptive advantages of using this method on green H₂ and syngas production. Two endothermic reactions, i.e., non-catalytic CH₄ pyrolysis and catalytic CH₄ dry reforming have been conducted as examples. As evidenced by the CFD simulations, the major innovation of our research is that the method enables rapid and uniform heating of high-throughput gaseous reactants to a precise reaction temperature. The method eliminates the need for a large number of elongated reactor tubes and external heating furnaces, leading to an energy-concentrated and ultra-compact reactor design potential over 2000 times smaller than industrial tubular reactor

systems with a side-fired furnace. The method has rarely been reported and thereby differentiates from the previous study. From the performance perspective, using just a piece of carbon, the distributed electrified heating method achieves CH₄ pyrolysis performance and stability beyond what many complex metal catalysts and ultra-high temperature heating technologies can do. For catalytic CH₄ dry reforming, the reactor with a washcoat of common Ni/MgO catalyst without any modification achieves exceptional activity performance at a syngas production capacity tens of times higher than other studies using precisely modified Ni/MgO-based catalysts. These great performances also differentiate from the previous study.

The state of the art is that, in modern chemical reactors, the energy supplied for heating reactants and reactions relies on the heat transfer from an external heated surface to the interior of reactants that flow. The low thermal conductivity of reactants (especially gaseous reactants) and the endothermic reactions lead to non-uniform temperature distribution, exhibiting significant temperature gradients across the reactor.

When preparing the manuscript, we realized that other prominent works have already been published on inductive-based heating for CH₄ valorization involving catalytic steam reforming or dry reforming. This is also the state of art of research. We have made a comparison between our work and the literature that uses induction heating and NiCo particles in Fig 3D (Reference 28-32). The common Ni/MgO catalyst used in this study outperforms the literature in terms of CH₄ conversion (Fig 3D-1), CO₂ conversion (Fig 3D-2), syngas production capacity (Fig 3D-3), and H₂ to CO ratio. It is clearly stated in the manuscript that: *“The literature’s induction heating method failed to perform well possibly due to the lack of regular micron-scale channels for uniform and rapid heating and immediate energy supply.”* Please refer to the highlighted content on page 9 in manuscript. The absence of regular micron-scale channels is a common issue in published inductive-heated research as they utilize either tubular reactors or pelletized catalysts. Publications (released more recently) that integrate the use of a structured catalyst

with direct electrical heating are considered to be more aligned with our concept, as noted in references 19 and 20 (<https://doi.org/10.1016/j.renene.2023.04.082>; <https://doi.org/10.1016/j.ijhydene.2022.12.346>). In their research, a clear explanation of the novelty of the method in terms of reactant heating and energy supply, which is verified by CFD simulations in our work, is missing.

Line 32: You shall replace “conversation” to “conversion”.

Response:

Thank you for your feedback. We have made the necessary correction by replacing the word "conversation" with "conversion" throughout the entire manuscript.

Line 57-58: “Though using a catalyst can lower this requirement...” ◊ Which requirement are you referring to?

Response:

Thank you for your feedback. The requirement refers to the temperature and the corresponding energy supply for the reaction. The content has been revised to be clearer. Please refer to the marked sentence on page 3 in the revised Introduction section.

Line 95-96: “...complete CH₄ decomposition at a constant temperature and corresponding chemical equilibrium (Fig 2A).” and Line 101: “...electrified heating are almost equivalent to the CH₄ equilibrium conversions” ◊ Could you please add the respective equilibrium figure for CH₄ pyrolysis for the temperature range you achieve in this work?

Response:

Thanks for the comment. The respective equilibrium for CH₄ pyrolysis has been added in the Supplementary Information as **figure S9**. Please refer to page 32 in Supplementary Information

file. Specific description of the method for the calculation of the thermodynamic equilibrium refer to Supplementary Information, page 18, marked with red-color:

"The thermodynamic equilibrium computations were performed employing a Gibbs free energy minimization method, utilizing Aspen Plus V12.1 software package. The selected equation of state was the Soave–Redlich–Kwong (SRK) model. This model asserts that all thermodynamic properties, including chemical potential, are determined based on the relationship between pressure, volume, and temperature of pure components and mixtures".

Line 101-102: "This indicates that the method enables all reactants to rapidly reach equilibrium..." ◇ Why do you use plural (reactants)? There is only CH₄ fed into the reactor.

Response:

Thanks for the comment. The original intention was to describe that all methane molecules can quickly reach equilibrium through this distributed electrified heating. The corresponding sentence has been revised in the manuscript, see page 5, line 113-114, with red color marked

Line 104: You must refer to Figure S7. Please replace Figure S5 to Figure S7.

Response:

Thank you for your comment. Figure S5 represents the schematic diagram of the induction heating furnace system that we included on line 107 as evidence of the use of the induction heating method. To provide further clarity, we have added testing performance data displayed in Figures S6 and S7. You may refer to the highlighted context on page 16 for more information.

Line 104-106: "Despite a significant increase in gas flux, the maintenance of CH₄ conversion at equilibrium conversion (100%) is ensured, highlighting the distinct benefits of using distributed electrified heating for high-throughput gas feeding." ◇ This is true only for the

highest temperature, i.e., 1150°C. Why is this not pronounced at lower temperatures (Figure S6)? Please present the equilibrium conversion for this temperature range.

Response:

Thank you for your comment. In our investigation of CH₄ pyrolysis, we first studied the effect of reaction temperature using a fixed gas flux of 750 h⁻¹ (GHSV). We examined temperatures of 850 °C, 950 °C, 1050 °C, and 1150 °C (see Figure S6). Next, we investigated the impact of GHSV on CH₄ conversion at a fixed temperature of 1150 °C, using elevated gas fluxes of 750 h⁻¹, 1500 h⁻¹, 3000 h⁻¹, and 6000 h⁻¹ (see Figure S7). Our findings show that despite the considerable increase in gas flux, the maintenance of CH₄ conversion at equilibrium (100% conversion) is ensured at 1150 °C. The influence of GHSV on CH₄ conversion at lower temperatures is not studied. Notably, it is possible to maintain the temperature of all reactants with increased gas flux by using distributed electrified heating. This indicates the maintenance of the reaction rate for all CH₄ reactants.

Line 107-108: “H₂ is the only detected gas compound (H₂ concentration of 100%).” ♦ At temperatures lower than 1150°C, other species (mainly C₂) are expected to be formed based on equilibrium composition presented in [doi.org/10.1016/0378-3820\(94\)00109-7](https://doi.org/10.1016/0378-3820(94)00109-7). Why do you detect only H₂ and carbon?

Response:

Thanks for your comment. The equilibrium composition is calculated ideally without considering subsequent reactions of the products. In our study, the temperature starts at 850 °C, which is high enough for the further decomposition of the species such as C₂H₄ (non-catalytic decomposition at a temperature higher than 730°C,

[https://www.researchgate.net/post/Does anyone know the thermal decomposition temperature of C₂H₄ without catalysts](https://www.researchgate.net/post/Does_anyone_know_the_thermal_decomposition_temperature_of_C2H4_without_catalysts)) and C₆H₆ (non-catalytic decomposition at a temperature

higher than 350 °C, [DOI 10.1143/JJAP.46.6037](https://doi.org/10.1143/JJAP.46.6037)) listed in the equilibrium composition. As mentioned in the manuscript, our distributed heating method has the major advantage of rapidly and uniformly heating reactants to a fixed temperature, which is then maintained. It means that although species such as C₂H₄ and C₆H₆ are produced, they will eventually be decomposed. It has been reported that maintaining a constant temperature results in the deep decomposition of CH₄ for coke and H₂ production, since intermediate species such as C₂H₄ and C₆H₆ (<https://www.nature.com/articles/s41586-022-04568-6>), once formed, will be decomposed immediately.

Line 127-128: "... which is 3 times higher than the capacity for H₂ production." ◇ This is redundant; C=12 g/mol and H=1 g/mol ◇ based on the mass balance, the mass production of C will always be 4 times the production of H (12 vs 4*1 in CH₄ molecule).

Response:

Thank you for the comment. The description here was to indicate the high conversion of CH₄ into both H₂ and carbon products. The sentence has been revised to make it more meaningful. See Page 6, lines 137-139.

Line 144-145: "Even though the wood carbon monolithic reactor can function reliably for a certain period, they will eventually reach the end of their life cycle." ◇ What is happening to its composition and structure? So, a continuous process (according to the string definition) is not feasible! A cyclic operating mode is rather feasible. Please be more precise.

Response:

Thanks for the comment. The author's point in this description is that carbon monolith, like the catalyst, has a specific lifetime as a reactor. However, even after it loses function, it can still be used as a promising carbon material in batteries. In this work, there was no observable functional decline or loss of the carbon monolith reactor, making it difficult to determine which

specific property changes led to the end of its lifetime. Moreover, the process is considered stable as a 1200-minute continuous test has been conducted with no obvious activity loss being observed.

Line 148: SIBs ◊ Please explain the acronym.

Response:

Thanks for the comment. We have added the full name, sodium ion batteries, when first introducing the abbreviation SIBs. See the highlighted content on page 7 in manuscript.

Line 165-167: “WHSVs). When the WHSV is 75.8 L gcat⁻¹ h⁻¹, a conversion rate of almost 100% is observed for both CH₄ and CO₂ at temperatures of 750 oC, 800 oC, and 850 oC.” ◊ As previously asked, please present the equilibrium performance of dry methane reforming at this temperature range.

Response:

Thanks for your comment. The equilibrium curve of dry CH₄ dry reforming is shown in Figure S21 in the updated Supplementary Information, page 44.

Line 172-173: “...a conversion rate of 100% for CO₂ is still maintained, while a conversion rate of approximately 90% is observed for CH₄.” ◊ 100% seems quite a lot. Please present the carbon lack in the conversion/temperature plots.

Response:

Thanks for your comment. In the section "**1.2.3 CH₄ and CO₂ conversion, syngas production capacity, H₂, and CO yield calculation**" of the Supplementary Information, you can find a list of possible reactions that may have occurred during the test. It is worth noting that there was minimal carbon deposition during the test and the CO yield was consistently higher than the H₂ yield. This suggests that the primary side reaction was the reverse water gas shift reaction,

resulting in H₂O being the only other product aside from H₂ and CO. This is consistent with the equilibrium figure shown in Figure S21. The yield of H₂O throughout the test has been included in Figure 3C.

Line 182-185: “As shown in Figure 3D, the common Ni/MgO catalyst used in this study outperforms most optimized catalysts in terms of CH₄ conversion (Fig 3D-1), CO₂ conversion (Fig 3D-2), syngas production capacity (Fig 3D-3), and H₂ to CO ratio (Fig 3D-4) with the assistance of distributed electrified heating, even without any modification.” ◊ Why? What is done differently than other works? It is not clear to me what the difference is; is the design of the reactor, the type of the catalytic material, the structure of the catalytic material or something else?

Response:

Thank you for your comment. As stated previously, the aim of this study is to present a method for distributed electrified heating and to demonstrate its benefits by utilizing two reactions - CH₄ pyrolysis and CH₄ dry reforming. The key difference from other works is the implementation of the distributed electrified heating method, which is derived from the reactor's design. The distributed electrified heating allows for fast and uniform heating of reactants to a specific temperature, which is not feasible with the traditional heating method. As a result, a common Ni/MgO catalyst used in this study, which is commonly used, outperforms most optimized catalysts in terms of CH₄ conversion (as shown in Fig 3D-1), CO₂ conversion (as shown in Fig 3D-2), syngas production capacity (as shown in Fig 3D-3), and H₂ to CO ratio (as shown in Fig 3D-4).

Line 188: “...indicating that side reactions are...” ◊ Why are those side reactions? Could you also comment on the reaction mechanism?

Response:

Thanks for the comment. As stated previously, a list of possible reactions that may have occurred during the test can be found in the section "**1.2.3 CH₄ and CO₂ conversion, syngas production capacity, H₂, and CO yield calculation**" in Supplementary Information. The reaction mechanism is similar to other reported research using Ni/MgO catalyst, where Ni aids in cracking and reforming reactions, and MgO is effective in activating CO₂ and resisting coke formation ([10.1126/science.aav2412](https://doi.org/10.1126/science.aav2412)). The major advantage of the current study is the rapid and uniform heating and the immediate energy supply for the reactions, resulting in the occurrence of reactions at a constant chemical equilibrium.

Line 191-192: "A higher syngas production capacity of a reactor means a higher reactant throughput..." ◊ Not necessarily; it can also attributed to higher single-pass conversion. Please rephrase accordingly.

Response:

Thanks for the comment. To avoid any misunderstandings, we have revised the sentence. See highlighted content in lines 205-207: "*When the conversion rate and product selectivity of a reaction are comparable, a higher syngas production capacity of a reactor means a higher reactant throughput and a correspondingly higher energy supply requirement.*"

Line 193-195: "The literature's induction heating method failed to perform well possibly due to the lack of regular micron scale channels for uniform and rapid heating and immediate energy supply." ◊ Please add the respective reference(s).

Response:

Thanks for your comment. Please refer to page 9, line 210 in the updated manuscript. A corresponding reference has been added (<https://doi.org/10.1016/j.ijhydene.2019.02.055>).

Line 200-201: "... is the way forward for large-scale application of the catalytic CH₄ dry reforming reaction." ◇ This is generality. The authors do not explain how their reactor can be scaled. Please elaborate further.

Response:

Thanks for your comment. To increase the capacity of our reactor, the most effective method is to demonstrate a larger reactor with the same geometry and install an appropriate induction heating furnace. The production of a bigger metal monolith is already possible, and high-powered induction heating furnaces are also available for industrial use. The main challenge lies in determining the dimensions of the monolithic reactor and the furnace's capacity based on the specific syngas production requirements. Moving forward, CFD simulations with larger gas feeding capacity is necessary.

Line 228-230: "In the distributed electrified heating scenario, the temperature gradient in the radial direction throughout the reactor is consistently close to 0, which aligns with its flat temperature contours." ◇ In my opinion, this is promoted by the small and confine geometry. What if the channels were wider, which might be the case for industrial scale reactors.

Response:

Thanks for your comment. The diameter of the micron-scale channel plays a crucial role in the heating of reactants. When the channels are wider, the heating of reactants in the middle is slower, which leads to a less uniform temperature distribution. This issue is a significant hurdle for the current commercial tubular reactor, which has a diameter of approximately 10 cm but a length over 10 m. Optimization of the geometry design using CFD simulations that maintain the micro-scale channels and low-pressure drop at high gas flux is recognized as essential for industrial reactors. Moving forward, this will continue to be a part of our work.

Line 237-238: “Perhaps the steady progress of CH₄ pyrolysis can be explained by this.” ◊ What would be the reactor performance if conductive (Joule) heating would be applied? I believe the result would be the same. If so, does inductive heating bring an added value as compared to other electrified heating methods such as conductive heating? Could you please comment on this matter?

Response:

Thanks for the comment. When it comes to producing H₂ and converting CH₄, the impact of using Joule heating versus induction heating is nearly identical. Both methods can realize uniformly heating of the reactants and immediate energy supply for the reactions. However, as stated in **Supplementary Discussion 2**, the carbons produced could potentially be heated by induction, leading to an autocatalytic effect to promote CH₄ conversion. This effect has also been observed in microwave-related research, which may further facilitate the formation of carbon nanofibers (<https://doi.org/10.1016/j.ijhydene.2022.12.353>), as observed in the current study. For Joule-heating-based study, the effect has not been reported. Therefore, in terms of carbon production, induction heating may differ from Joule heating.

Line 267-268: “This enables the energy to be used for gas heating and endothermic reactions to a greater extent, conforming to a highly energy- concentrated and ultra-compact system.” ◊ What is the global energy efficiency (electricity-to-chemical energy)?

Thanks for your comment. The author from the Technical University of Denmark has systematically studied the energy efficiency of converting electricity to chemical energy using induction heating (<https://doi.org/10.1016/j.cattod.2019.05.005>). In general, the reactor's efficiency increases with the capacity of its H₂ production efficiency. A technical and economic analysis of the CH₄ pyrolysis process was conducted using 100 kg/h of natural gas. An energy efficiency value of approximately 88.3% was used for the analysis, which resulted in an energy

requirement of about 1.28 kWh/Nm³-H₂ for H₂ production. The corresponding content has been added into the section **Techno-economic analysis** in manuscript. See page 12-13, lines 297-300:

“ An energy efficiency value of approximately 88.3% was used for the analysis. The specific energy consumption for H₂ production is calculated to be approximately 1.28 kWh/Nm³-H₂. ”

Supplementary information material

Figure S6 ◊ Why does CH₄ conversion decrease over the time at 850, 950 and 1050 oC while at 1150 oC remains stable?

Response:

Thanks for the comment. As shown in Figure S6, CH₄ conversions at lower temperatures are not as stable as those at 1150 °C. It is more accurate to describe a fluctuated tendency rather than a decreased one for CH₄ conversions at lower temperatures. As stated in the manuscript, temperature is crucial for the conversion of CH₄. It determines the reaction rates as well as the corresponding chemical equilibrium. The major reason for the fluctuation in CH₄ conversion at lower temperatures is believed to be the temperature fluctuation caused by the induction heating furnace. This has also been observed in the stability test shown in Figure 2B.

Table S3 ◊ I strongly recommend you to include an additional metric, that is the energy cost: actual energy provided vs theoretical energy needed to attain the same result for each case. This will allow fair comparison considering all important aspects.

Response:

Thanks for your comment. As stated previously, a techno-economic analysis of the CH₄ pyrolysis process based on the experimental data has been conducted and the results are added to the manuscript. The energy cost regarding the H₂ production has been calculated to be around

1.28 kWh/m³. We have compared it with the lowest specific energy consumption values for H₂ production in other research that uses electrified technologies. Our findings show that the specific energy consumption for H₂ production in this work is comparable, and in some cases even much lower than other electrified heating technologies.

The authors have acknowledged that a fairer comparison would be to take into account the energy cost. Unfortunately, there is a lack of available information for reference in this regard. We have listed all the published information we could find from literature and websites in Table S3. Moreover, it's worth noting that the energy costs for lab or demonstration scale tests are significantly higher compared to commercial scale plants.

Reviewer #3 (Remarks to the Author):

The work is interesting and useful and has the potential to be accepted, but some points must be clarified or fixed before I can proceed and positive action can be taken.

1- The abstract is long and should be more concise.

Response:

Thanks for the comment. We have revised the abstract section by removing the background information and reference literature to make it more concise. Please refer to page 2 to review the updated **Abstract** section.

2- Why does the paper lack an Introduction section and a review of previous relevant studies? Also, there is no conclusion section.

Response:

Thanks for the comment. The structure of the paper has been revised accordingly. Sections including “Introduction”, “Conclusions”, and “Methods” have been added. Additionally, we have included a brief overview of past studies that are relevant to our research. To support our

claims, we have also incorporated recent research papers. Please refer to page 3-4, paragraph 2 to review the updated content in the Introduction section.

3- In line 278, it is mentioned that 3000 h-1 represents an industrial GHSV. The reference of this issue must be addressed.

Response:

Thank you for your comment. Our statement was declared based on a report from a demonstration project titled “High-Throughput Methane Pyrolysis For Low-Cost, Emissions-Free Hydrogen”, which was funded by the "U.S. Department of Energy Hydrogen Program Plan" (reference 23 in the manuscript, <https://arpa-e.energy.gov/sites/default/files/7%20H2%20Performers%20PARC.pdf>). After considering your feedback, we realized that using the term "industrial-level" was not appropriate as the technology is still in the demonstration phase. Therefore, we have made the necessary revisions and replaced "industrial-level" with "high-throughput". Please review the highlighted content on page 6, line 129, and page 15, line 358.

4- According to the paper, mini reformers such as compact monolithic modules are good candidates to apply the approach of distributed electrical heating. This is while, the use of small-scale reactors that include micro-channels, on an industrial scale is associated with limitations. This problem is aggravated in the case of reactions that cause path closure by producing coke (such as methane pyrolysis); because decoking of micro-channels is a tedious attempt. This reduces the industrial attractiveness of these reactors and casts doubt on the authors' claim about the potential of creating an industrial revolution. If the authors believe that this method is only applicable to micro-scale reactors, they should clearly explain the limitations of using this method in industrial scales.

Response:

We appreciate your comment and fully agree with your point about the path closure caused by

carbon production in the channels hindering the industrial application of compact monolithic modules with micro-scale channels. However, we have conducted a CH₄ pyrolysis test for 1200 minutes at 1150 °C and GHSV of 3000 h⁻¹ using an 8 cm³ wood carbon monolith. Despite producing approximately 257g of carbon during the test, we did not detect any channel closure and achieved complete CH₄ conversion along the test (Figure 2B).

Moreover, we have realized that this issue when preparing the manuscript and have made some discussion in the manuscript, which is listed in the section **Supplementary Discussion 3, Rapid movement of fibrous carbon with H₂ flow** (pages 22 in Supplementary Information).

The text is copied below for your clearance:

“One of the biggest obstacles in implementing this technology is the risk of micron-scale carbon channel blockages caused by fibrous carbons. These blockages can cause the reaction to shut down. However, it is confirmed that solid carbons can be quickly carried out from the carbon channels during testing, ensuring a continuous process described previously. Most of the fibrous carbons are located on the top of the carbon monolith (free falling by gravity) and in the top cooling area of the connecting flanges after the test (figure S14). The channel size, which ranges from 10-60 μm, is notably larger than the bulk size of the fibrous carbon, which is less than 1 μm. As a result, most solid carbons are carried with the gas flow, with only a small portion attaching to the walls of the channel (figure S15). When comparing the fresh carbon monolith to the spent one, the reduction in specific surface area is a clear indication of secondary pores (on channel walls) closure by deposition of carbon (table S1). The leading cause of deactivation in activated carbon catalysts for CH₄ pyrolysis has been reported as the closure of the inner pores within the same range as the secondary pores on the channel wall. In this study, the closure of these pores on the channel does not appear to result in a decrease in the CH₄ conversion rate, as the main channels remain unimpeded (figure S15).”

Additionally, in the CFD simulations section, the gas velocity calculation result shows that *“the gas velocity in the distributed electrified heating scenario (up to 0.7 m/s, figure S24) is significantly higher than in the external heating scenario (up to 0.16 m/s, figure S25) due to the rapid heating and uniform temperature distribution.”* (Manuscript, lines 247-250, page 10). It can be seen that the movement of carbon products in micron-scale channels is facilitated by our method as the speed of gases is much higher. The distributed electrified heating method could maintain a similar reaction situation in each channel, regardless of the size of the monolith. Our results suggest that the use of the distributed electrified heating method can significantly reduce the risk of path closure. Therefore, we believe that our method can be applied not only to micro-scale reactors but also has significant potential for large-scale applications.

REVIEWERS' COMMENTS

Reviewer #1 (Remarks to the Author):

In my opinion the authors improved the manuscript, which can be considered for publication.

Reviewer #2 (Remarks to the Author):

The authors have sufficiently addressed all my technical comments.

However, it is still not clear to me what the novelty of this reactor is compared to other electrified reactors (i.e. Joule heated reactors) which also feature distributed electrified heating.

The response of the authors to my respective comment (Line 182-185: "As shown in Figure...any modification."  Why? What is done differently than other works? It is not clear to me what the difference is; is the design of the reactor, the type of the catalytic material, the structure of the catalytic material or something else?) refers to conventional fire-heated systems but not to the electrified ones.

Could the authors elaborate further on this matter? Could the authors mention the differences between their inductively heated system and Joule heated reactors?

In Figure 6, the performance of those two systems is rather comparable (if not the same).

Reviewer #3 (Remarks to the Author):

The article is now proper to published in the journal of Nature Communications.

Reviewer #1 (Remarks to the Author):

In my opinion the authors improved the manuscript, which can be considered for publication.

Reviewer #2 (Remarks to the Author):

The authors have sufficiently addressed all my technical comments.

However, it is still not clear to me what the novelty of this reactor is compared to other electrified reactors (i.e. Joule heated reactors) which also feature distributed electrified heating.

The response of the authors to my respective comment (Line 182-185: “As shown in Figure...any modification.”  Why? What is done differently than other works? It is not clear to me what the difference is; is the design of the reactor, the type of the catalytic material, the structure of the catalytic material or something else?) refers to conventional fire-heated systems but not to the electrified ones.

Could the authors elaborate further on this matter? Could the authors mention the differences between their inductively heated system and Joule heated reactors?

In Figure 6, the performance of those two systems is rather comparable (if not the same).

Response:

Thank you for this valuable question.

In the previous version of the response, we explained the advantages of our distributed electrified heating method compared to traditional heating methods when used to absorb heat in chemical processes. In section **Efficient catalytic CH₄ dry reforming with high syngas production capacity**, we compared our own experimental results with literatures implementing optimized catalysts or advanced heating methods. We also clarified the reason why our proposed method outperformed most of those in the literatures even using induction heating. It is because we applying ditributed monolith-type reactor via induction heating, realizing a real uniform heating.

It is true that this distributed electrified heating method can be achieved via both Joule heating and Induction heating. Both Joule heating and Induction heating are pioneering heating methods raised recently applying in chemical engineering. Joule heating, as known as resistive heating,

generates heat when an electric current passes through a conductor due to the material's electrical resistance. Induction heating generates heat by inducing eddy currents in the materials through electromagnetic induction. The key differences include: 1). Joule heating requires direct contact with the power source, whereas induction heating is a non-contact method. 2). Joule heating is effective for any conductive material, while induction heating is most efficient for materials with good electrical conductivity and magnetic properties.

3). Joule heating can lead to uneven heating in materials with variable resistance, while induction heating can provide more localized heating.

For the specific technique reason between Induction-heating and Joule-heating in the studied case, as we explained in your former comment (*Line 237-238: "Perhaps the steady progress of CH₄ pyrolysis can be explained by this."* □ *What would be the reactor performance if conductive (Joule) heating would be applied? I believe the result would be the same. If so, does inductive heating bring an added value as compared to other electrified heating methods such as conductive heating? Could you please comment on this matter?*), both heating methods can realize fast and uniform heating of the reactants and immediate energy supply for the reactions via our monolith reactor. In addition, the concentrated localized heating effect of the induction heating facilitates fibrous carbons production instead of carbon black and the newly produced carbon from methane pyrolysis could be induction-heated, which cannot be realized by Joule-heating.

The manuscript and the supplementary information have been revised to clarify this main point, marked in red color.

Reviewer #3 (Remarks to the Author):

The article is now proper to published in the journal of Nature Communications.